# Usefulness of the ECORE-BF Scale to Determine Atherogenic Risk in 386,924 Spanish Workers

**DOI:** 10.3390/nu16152434

**Published:** 2024-07-26

**Authors:** Marta Marina Arroyo, Ignacio Ramírez Gallegos, Ángel Arturo López-González, María Teófila Vicente-Herrero, Daniela Vallejos, Tomás Sastre-Alzamora, José Ignacio Ramírez Manent

**Affiliations:** 1Research ADEMA SALUD Group, University Institute for Research in Health Sciences (IUNICS), 07010 Palma, Balearic Islands, Spain; mmarinaa@its.jnj.com (M.M.A.); ignacioramirezgallegos@gmail.com (I.R.G.); correoteo@gmail.com (M.T.V.-H.); d.vallejos@eua.edu.es (D.V.); tsastre04@sonrie.com (T.S.-A.); joseignacio.ramirez@ibsalut.es (J.I.R.M.); 2Faculty of Dentistry, ADEMA University School, 07010 Palma, Balearic Islands, Spain; 3Institut d’Investigació Sanitària de les Illes Balears (IDISBA), Health Research Institute of the Balearic Islands, 07010 Palma, Balearic Islands, Spain; 4Health Service of the Balearic Islands, 07010 Palma, Balearic Islands, Spain; 5Faculty of Medicine, University of the Balearic Islands, 07010 Palma, Balearic Islands, Spain

**Keywords:** obesity, atherosclerosis, body fat, atherogenic index, ECORE-BF

## Abstract

Background: Cardiovascular diseases are the leading cause of death worldwide. Obesity and atherosclerosis are considered risk factors for this pathology. There are multiple methods to evaluate obesity, in the same way as there are different formulas to determine atherogenic risk. Since both pathologies are closely related, the objective of our work was to evaluate whether the ECORE-BF scale is capable of predicting atherogenic risk. Methods: Observational, descriptive, and cross-sectional study in which 386,924 workers from several autonomous communities in Spain participated. The association between the ECORE-BF scale and five atherogenic risk indices was evaluated. The relationship between variables was assessed using the chi-square test and Student’s *t* test in independent samples. Multivariate analysis was performed with the multinomial logistic regression test, calculating the odds ratio and 95% confidence intervals, with the Hosmer–Lemeshow goodness-of-fit test. ROC curves established the cut-off points for moderate and high vascular age and determined the Youden index. Results: The mean values of the ECORE-BF scale were higher in individuals with atherogenic dyslipidemia and the lipid triad, as well as in those with elevated values of the three atherogenic indices studied, with *p* <0.001 in all cases. As atherogenic risk increased across the five evaluated scales, the prevalence of obesity also significantly increased, with *p* <0.001 in all cases. In the ROC curve analysis, the AUCs for atherogenic dyslipidemia and the lipid triad were above 0.75, indicating a good association between these scales and the ECORE-BF. Although the Youden indices were not exceedingly high, they were around 0.5. Conclusions: There is a good association between atherogenic risk scales, atherogenic dyslipidemia, and lipid triad, and the ECORE-BF scale. The ECORE-BF scale can be a useful and quick tool to evaluate atherogenic risk in primary care and occupational medicine consultations without the need for blood tests.

## 1. Introduction

Overweight is considered an important risk factor for cardiovascular diseases, including atherosclerosis [1]. There are different methods to measure excess weight, whether obesity or overweight; some focus only on total weight, while others place greater emphasis on excess body fat [2]. One of the most used formulas is the body mass index (BMI), which has been very useful in epidemiological studies. However, this formula does not inform us about body composition or the amount of fat and muscle mass of the person being evaluated. When determining the relationship between an individual’s weight and height, it does not differentiate between fat mass and lean mass. Therefore, people who play sports and have a high amount of muscle tissue will have a high BMI without having excess fatty tissue [3].

It is currently accepted that it is the accumulation of excessive fat that is harmful to health, especially so-called visceral fat [4,5,6]. Various methods are available to determine the amount of body fat. Among these we find the determination of skin folds, in which the thickness of subcutaneous adipose tissue is determined in various areas of the body, established using a constant pressure lipocaliber of 10 gr/mm^2^. From these measurements, body fat is estimated using validated regression equations [7,8]. Another of the techniques used is bioelectrical impedance (BIA), which is a non-invasive method that enables body composition to be evaluated. This technique is based on the electrical properties of body tissues, which make it possible to evaluate fat mass, fat-free mass, skeletal muscle mass, bone minerals, total body water, and degree of hydration [9,10]. 

At this time, bone densitometry is considered the gold standard for determining fat mass [11]. There are different techniques to calculate the density of different parts of the body. Dual X-ray absorptiometry (DEXA) is the tool of choice according to the European and American nutrition societies to assess body composition. It produces low radiation and has good precision in measuring the lean mass, fat mass, and bone mass compartments [12,13]. Other techniques used are air displacement plestymography using the POD BOD (computerized device with an egg-shaped fiberglass chamber) which uses the relationship between pressure and volume to determine the body volume of the person sitting inside the chamber. Measuring body weight and volume enables the calculation of body density and estimation of the percentage of fat and fat-free mass [14,15]. Hydrodensitometry has also been used to determine body composition. This measurement is carried out by immersing the subject in a tank full of water after forced expiration. Subsequently, following Archimedes’ principle, the body volume inside and outside the water is estimated by applying a correction factor according to the density of the water used. In this way, body density, fat mass, and lean mass are obtained [16,17]. The standard technique requires complete immersion of the individual, which can cause anxiety or aspiration of water, among other complications. Thus, the procedure has been updated by carrying out measurements with the head out of the water [18,19].

All of these are laboratory techniques to evaluate body composition; however, they are not suitable for use in field work as the tools are not widely available, and they involve transportation of participants to tertiary care centers.

Alternatively, we have different body fat estimators. These formulas evaluate anthropometric parameters (height, and hip, waist, and neck circumference measurements) and sex to obtain an estimate of the individual’s body fat percentage.

Among the body fat estimators, the most used is the CUNBAE (body adiposity estimator of the Navarra University Clinic), which was validated a few years ago [20]. However, since it is a complicated formula, a group of researchers from Córdoba developed another simpler formula called ECORE-BF (Córdoba body fat equation). This new formulation, much more straightforward when compared with the CUNBAE, showed a correlation of 0.998 in more than 196,000 people [21].

Atherosclerosis is a multifocal immunoinflammatory disease that causes the accumulation of plaques of fat, cholesterol, and other materials on the walls of medium and large-caliber arteries, which affects them in the long term [22]. This condition increases the risk of cardiovascular disease by promoting stiffness and rupture of arteries [23].

Atherosclerosis and obesity are closely related and both have detrimental effects on cardiovascular health [24]. The incidence of cardiovascular diseases related to obesity has increased worldwide, and constitutes a true public health problem. Thus, obesity increases cardiovascular risk through several elements associated with atherosclerosis, modifications in adipokines, inflammation, oxidative stress, and others, which produce endothelial dysfunction [25]. Obesity is considered a disease linked to lifestyle since its incidence is influenced by diet, sedentary lifestyle, stress, smoking, and excessive alcohol consumption [26,27].

There appears to be a strong association between obesity and atherosclerosis, which could lead to an increase in the occurrence of cardiovascular diseases [28]. In order to reduce these risks, it is essential to identify atherogenic risk and maintain a healthy lifestyle.

The aim of our work was to evaluate whether the ECORE-BF scale, which is easy to determine in primary care and occupational medicine consultations, is capable of predicting atherogenic risk; in such a way, people with a high atherogenic risk would be identified early and quickly, in order to develop the appropriate measures to make changes that modify it. These measurement and correction measures would be easily implemented in primary care and occupational medicine consultations, so as to reduce cardiovascular events (stroke and heart disease) and improve the effectiveness of the health system and the quality of life of the population.

## 2. Materials and Methods

### 2.1. Participants

This article is based on an observational, descriptive, and cross-sectional study in which 386,924 workers from several autonomous communities in Spain participated. Men made up 60.2% of the population (232,814), while women made up 39.8% (154,110). The employees included in this study are those who showed up for the recurring medical examinations held by each of the participating companies. The time frame for the study ran from January 2019 to June 2020.

The inclusion criteria to select the sample are detailed below:-Being in the age range of 18 to 69.-Holding a job contract with one of the participating businesses.-Agreeing to participate in the research.-Agreeing to the transfer of data used for epidemiological research.

The exclusion criteria were as follows:-Age under 18 years or over 69 years.-Not being an employee of one of the participating companies.-Refusal to participate in the research study.-Refusal to consent to the use of data for epidemiological research.

Figure 1 shows the data flow diagram of the workers after applying the inclusion criteria.

### 2.2. Determination of Variables

The individuals who work in the occupational health departments of the participating companies were in charge of gathering the information required for this investigation. One method used to gather the data was anamnesis. Data on sociodemographic factors (age, sex, socioeconomic class, and education level) and healthy habits (tobacco, alcohol, Mediterranean diet, and physical activity) were gathered through a thorough clinical history.
-Clinical and anthropometric computations included height, weight, waist circumference, and systolic and diastolic blood pressure.-Analytical conclusions, glycaemia, and lipid profiles were established.

Standardization of the variable measurement methodologies was an attempt to prevent potential biases in the study.

#### 2.2.1. Anthropometric Determinations

Subjects were measured for height and weight while standing, wearing only underpants, with their arms dangling and their head and thorax aligned. Values are given in kilos and millimeters and were measured using a SECA-type scale-measuring apparatus, following international standards for ISAK anthropometric evaluation [29].

Abdominal waist circumference was measured using a SECA model measuring tape parallel to the floor and positioned halfway between the last rib and the iliac crest. The person’s abdomen was relaxed as they stood. In this position, hip circumference was also measured by positioning the tape measure parallel to the floor at the level of the widest point of the buttocks [30,31].

#### 2.2.2. Clinical Determinations

An OMROM-M3 blood pressure monitor was used to measure blood pressure. The subject had to be seated and at rest for a minimum of ten minutes in order to perform an accurate assessment. There were cuffs in various sizes because the cuff was wrapped around the arm until it fitted snugly without being too tight. Three measurements were conducted in a row, one minute apart. The average of the three figures was the value assessed. Individuals were considered hypertensive if their systolic blood pressure was above 140 mm/Hg, diastolic blood pressure was above 90 mm/Hg, or if they were undergoing treatment for hypertension.

#### 2.2.3. Analytical Determinations

Venipuncture was used to extract blood, and it required a 12 h fast beforehand. For the purpose of optimal preservation, samples were processed and kept in a refrigerator for a maximum of 48 to 72 h and then analyzed in reference laboratories using comparable techniques. Enzymatic methods were used to measure blood glucose, triglycerides, and total cholesterol, while precipitation methods were used to measure HDL cholesterol [32,33]. The Friedewald formula was used to estimate LDL cholesterol indirectly, which is valid as long as the level of triglycerides in the blood is less than 400 mg/dL. LDL was measured directly if the value was greater than 400 mg/dL. The expression for each analytical variable is mg/dL. Patients were considered dyslipidemic if their lipid levels exceeded the reference laboratory cutoff points or if they were receiving treatment for dyslipidemia.

#### 2.2.4. Risk Scales

The Córdoba body fat equation (ECORE-BF) [34] is determined from this formula:

ECORE-BF = −97.102 + 0.123 (age) + 11.9 (gender) + 35.959 (LnBMI), where male = 0 female = 1.

Cut-off points are as follows: normal weight (<30 in women and <20 in men), overweight (30–35 in women and 20–25 in men), and obesity (>35 in women and >25 in men).

Atherogenic risk was estimated by applying different formulas:

High triglyceride levels, low HDL cholesterol (less than 50 mg/dL in women and less than 40 mg/dL in men), and normal LDL cholesterol values were used to diagnose atherogenic dyslipidemia. When LDL cholesterol levels surpassed 160 mg/dL, a lipid triad was taken into consideration [35]. 

Different cut-off values apply to different atherogenic indices: 

Total cholesterol/HDL-c index—low risk: less than five for men and less than four for women; moderate risk: five to nine for men and four to seven for women; high risk: greater than nine for men and greater than seven for women.

For the LDL-c/HDL-c ratio—low risk: <3; and high risk: >3. 

The triglycerides/HDL-c ratio is regarded as high risk from 3% [36].

Male and female were the two dichotomous variables making up gender. 

The date of the medical examination was subtracted from the date of birth to determine age. 

The educational level taken into consideration was the highest of all those completed. Primary studies, secondary studies, and university studies were the three levels defined.

By applying the Spanish Society of Epidemiology’s guidelines and taking into account the kinds of jobs included in the 2011 national classification of occupations (CNO-11) [37], social class was ascertained. There are three established levels: -Social class I: This comprises university-educated professionals, managers, professional athletes, and artists.-Social class II: This is made up of intermediate professions and qualified self-employed people.-Social class III: This group consists of workers with less qualification.

In this study, someone was deemed to be a smoker if they had not stopped smoking for less than a year beforehand or if they had used tobacco products at least once a day during the previous 30 days.

Adherence to the Mediterranean diet was evaluated using a questionnaire that had 14 questions with a value of 0 or 1 point. Nine or more points show a high level of adhesion [38]. The International Physical Activity Questionnaire (IPAQ) was used to measure the degree of physical activity. This self-administered survey measures how much exercise has been done during the previous seven days [39].

### 2.3. Statistical Analysis

Quantitative data were expressed using mean and standard deviation (SD), while categorical variables were reported using frequency and percentage. The chi-squared test and Student’s *t*-test were used to assess whether there were statistically significant differences between the different variables. Previously, validation statistics were used to check that the samples followed a distribution that did not allow us to use these parametric tests. 

ROC curves were used to construct cut-off points for moderate and high vascular age values, as well as to determine sensitivity, specificity, and Youden index. SPSS 28.0 software was utilized for statistical computations, and a significance level of *p* < 0.05 was established.

## 3. Results

Table 1 displays the anthropometric, clinical, analytical, sociodemographic, and healthy habit data for the 386,924 study participants. The participants’ average age was just over 39 years old. The variables in the male group exhibited larger negative values, with the exception of LDL cholesterol. Men made up 60.2% of the total, while women made up 39.8%. Most people in the population were within the age range of 30 to 49. The majority were from socioeconomic class III and had only completed primary school education. Of the men, 45.5% frequently engaged in physical exercise whereas it was 52.2% of women, while 41% of men and 51.4% of women adhered strictly to the Mediterranean diet. There was a smoking habit among 33% of women and just over 37% of men.

Table 2 shows that the mean values of the ECORE-BF scale were higher in people with atherogenic dyslipidemia and lipid triad and in those with higher values of the three atherogenic indices studied. In all cases, these ECORE-BF values were higher in women.

Something similar occurred with the prevalence of obesity determined with ECORE-BF; that is, this prevalence was greater in people with atherogenic dyslipidemia, lipid triad, and high values of the different atherogenic indexes. However, the prevalence of obesity was greater in men except when atherogenic dyslipidemia was assessed. The complete data can be consulted in Table 3.

Figure 2 and Figure 3 and Table 4 show the different ROC curves, in both men and women, in which it can be observed that the highest areas under the curve (AUCs) are seen for atherogenic and lipid triad in both sexes, with values around 0.8, which can be considered high. The AUC for the atherogenic indices were lower. Except for total cholesterol/HDL-c, the rest of the AUCs were higher in men.

Table 4 shows the different ECORE-BF cut-off points for determining the occurrence of AD, LT, and high values of the different atherogenic indices.

Table 5 shows the values of Cohen’s kappa concordance indexes for the different indicators of atherogenesis analyzed, showing that in all cases they were very low.

## 4. Discussion

Obesity and cardiovascular diseases currently constitute a priority and growing problem worldwide, and their incidence has been increasing in recent years. The main factor of cardiovascular pathology is widely accepted to be atherosclerosis. In this study we establish the association between a formula to estimate body fat that is easy to apply and calculate in medical consultations (ECORE-BF), and five formulas for calculating atherogenic risk.

Today, obesity is defined as a metabolic disease of multifactorial origin with excess body fat, especially visceral fat, which is harmful to health [40]. This nutritional disorder is on the rise worldwide, and has become so worrying that the WHO recognizes it as a global pandemic [41,42]. Among the pathologies caused by obesity are cerebrovascular accidents [43,44], heart diseases [45,46], high blood pressure [47,48], type 2 diabetes (T2DM) [49,50], insulin resistance [51,52], and dyslipidemia [53,54]. 

Visceral fat produces a large amount of adipokines, which are mainly pro-inflammatory elements; in such a way that in obesity there is a decrease in anti-inflammatory adipokines (adiponectin), whereas there is an increase in pro-inflammatory ones such as resistin. The latter activates the synthesis of proinflammatory cytokines such as TNF-α, IL-1, IL-6, and IL-12 [55], which facilitate vascular inflammation [56]. In this way, resistin could contribute to the development of atherosclerosis caused by obesity [57]. Another component of obesity is oxidative stress, due to increased production of reactive oxygen species by adipose tissue, thereby creating an imbalance between free radicals and antioxidant substances [58,59], which also favors the development of arteriosclerosis [60]. The release of fatty acids by adipose tissue directly causes an alteration of the vascular endothelium [61]. This state of sustained chronic inflammation affects the endothelium of blood vessels [62,63] at both the macrovascular and microvascular levels [64], and stimulates the development of atherosclerosis. Diabetes, high blood pressure, insulin resistance, and other metabolic disorders such as dyslipidemia contribute to this vascular alteration [65]. 

Metabolic syndrome (Met-S) is one of the metabolic disorders associated with obesity. This syndrome encompasses various factors that increase cardiovascular risk. The clinical abnormalities that constitute Met-S vary depending on the criteria applied, but generally include elevated blood glucose levels, increased waist circumference, elevated blood pressure, low HDL cholesterol levels, elevated triglycerides, and insulin resistance [66]. Changes in lifestyle habits, including the abandonment of the Mediterranean diet and an increase in the consumption of industrial products (such as desserts, sweets, potato chips, crackers, sugary or carbonated drinks, etc.), along with more sedentary lifestyles, have contributed to an increase in obesity and MetS in the population [67]. This is influenced by a person’s habitat. In urban areas, where there are large supermarket chains, the population tends to abandon diets based on vegetables, fruits, and fresh products. Conversely, in rural areas, there is a tendency to consume seasonal vegetables and poultry, resembling the Mediterranean diet [68]. Several authors have demonstrated that the Mediterranean diet significantly reduces the risk of developing metabolic syndrome (MetS) [69,70,71,72]. This reinforces the importance of modifying harmful lifestyle habits to decrease cardiovascular pathology.

Type 2 diabetes mellitus contributes to the development of atherosclerosis, alters blood flow, and increases the inflammatory state. In recent years, telomere length has been associated with various diseases, including T2DM [73], cardiovascular diseases [74], and obesity [75], and is considered a marker of cellular aging. Ojeda-Rodríguez et al. [76] found that patients with coronary artery disease and a risk of short telomeres had a higher risk of developing T2DM. The same study observed an association between adherence to the Mediterranean diet and a reduced risk of developing T2DM in individuals with short telomeres, highlighting the importance of preventive action [76].

Obesity, T2DM, and MetS are the most common risk factors for cardiovascular diseases, and all are implicated in the pathogenesis of atherosclerosis. The Mediterranean diet has demonstrated its beneficial role in the prevention and treatment of these conditions [77,78]. Additionally, in recent years, gut microbiota has been linked to various cardiometabolic disorders such as atherosclerosis and insulin resistance [79]. This gut microbiota is influenced by an individual’s diet, and the Mediterranean diet promotes the development of gut flora that reduces the risk of developing obesity, insulin resistance, and MetS, among other diseases [80]. As mentioned earlier, these conditions promote the development of atherosclerosis, which is the ultimate cause of cardiovascular pathology.

Studies on gut microbiota have identified the same bacteria in atheromatous plaques that are found in the oral cavity and intestines, indicating that these bacteria colonize plaque [81]. This association between atheromatous plaques, gut microbiota, diet, and cardiovascular disease must be considered in the treatment and prevention of this pathology. It is estimated that changing from an unhealthy diet to a healthy one, such as the Mediterranean diet, could reduce cardiovascular diseases by around 50%. This underscores once again the importance of preventive measures [82].

Atherosclerosis begins early in life; however, it tends to remain asymptomatic for a long time, leading to what is known as subclinical atherosclerotic disease. In this stage, atherosclerotic plaques can be detected in the aortic, ileofemoral, or carotid regions [83]. Subclinical atherosclerosis is often a risk factor for cardiovascular events that is not typically detected in routine clinical practice and has a high prevalence in asymptomatic middle-aged individuals [84,85].

The origin and development of atherosclerosis involve numerous factors, including oxidative stress, inflammation, and modifications in adipokines, ultimately leading to endothelial dysfunction with LDL cholesterol infiltration and atheroma plaque formation. These processes are influenced by a range of modifiable and non-modifiable risk factors [86]. Healthy lifestyle habits are among the modifiable factors. In the ILLERA study, Rojo-López et al., [87] found an association between adherence to the Mediterranean diet and a lower prevalence of subclinical atherosclerosis. This finding supports the objective of our work, which is to detect individuals with high atherogenic risk early on to implement appropriate measures for lifestyle changes.

In our study, we evaluated a sample of 386,924 workers, from practically all labor sectors, who carry out their work activity in the different Spanish autonomous communities (Balearic Islands, Canary Islands, Andalusia, Valencian Community, Madrid, Catalonia, Castile and León, Castilla-La Mancha, and the Basque Country): 232,814 men (60.2%) and 154,110 women (39.8%). The average age of the individuals was around 39 years old. Most of them had primary education and belonged to socioeconomic class III. In our sample, smoking occurred in 33% of women and 37.1% of men. These data are particularly worrying, since they are higher than those obtained in the European Health Survey in Spain in 2020, where women smokers had a percentage of around 20% and there was a slightly higher percentage of men, at 30% [88]. This is even more alarming if we compare them with the data obtained by the INE (National Institute of Statistics) of Spain in 2022, where the national percentage of smokers was 17% [89]. It is striking that 45.5% of men and 52.2% of women do physical exercise regularly. This does not coincide with the data published by the INE (National Institute of Statistics) of Spain in 2022, where only 39.7% of men performed regular physical exercise, and this percentage was also lower in women, 35.7% [90]. This also contrasts with different publications worldwide where women report a lower level of regular physical exercise than men [91,92,93]. Regarding the clinical and analytical variables, the male group presented worse results in all of them except for LDL cholesterol, which was significantly higher in the female group. These results might be influenced by adherence to the Mediterranean diet, which is significantly higher in women: 51.4% compared to 41% in men [94,95].

When studying the mean values of the ECORE-BF scale, they were higher in people who have atherogenic dyslipidemia and lipid triad. In the other three atherogenic indices studied, we observed that the mean ECORE-BF values also increased in parallel and significantly as the value of the atherogenic risk index increased.

When evaluating the prevalence of obesity determined with ECORE-BF, we found that it was higher in the population with atherogenic dyslipidemia, lipid triad, and high values of the other three atherogenic indices evaluated. When comparing both results, it is striking that when studying the average results of the ECORE-BF scale, the highest atherogenic risk values were obtained in women with the five formulas used; whereas when assessing the prevalence of obesity, these results were reversed and were higher in men, except for atherogenic dyslipidemia, which presented a higher percentage in women.

Our results indicate that as atherogenic risk increases, the prevalence of obesity also increases, and this occurs for both sexes.

In our bibliographic search, we found no article that relates obesity determined by the ECORE-BF index to atherogenic risk indices. Hence, we proceeded to make comparisons with other studies that relate obesity, measured by another formula, to plasma atherogenic risk. The first of these studies was conducted in Taiwan by Zhang et al. [96]. In this work, the authors defined obesity using BMI and atherogenic risk using log10 (TG/HDL-C). Their results coincide with those obtained by us, in that people with an atherogenic index at higher levels have a greater risk of suffering from obesity. Another study carried out in China that used the same parameters as the previous one obtained identical results [97]. A Turkish study that used the same formulas as in the Chinese population, in patients with familial Mediterranean fever, concluded that for these patients there was a strong positive association between plasma atherogenic risk and obesity [98]. In Korea, Shin et al., [99], when evaluating plasma atherogenic risk using the log10 formula (TG/HDL-C), found that a low atherogenic risk was associated with a low BMI and a smaller waist circumference. In a prospective cohort study with 17,944 people also carried out in Korea, it was revealed that as plasma atherogenic risk increased, BMI increased at the same time [100].

In the analysis of the ROC curves for the five atherogenic risk formulas evaluated, the areas under the curve (AUC) with the best results correspond to atherogenic dyslipidemia, with a value of 0.803 in women and 0.812 in men; and to the lipid triad, with a value of 0.797 in women and 0.798 in men—in both cases, greater than 0.75, which allows these results to be deemed as good. Although Youden’s indices are not excessively high, they are around 0.5.

When analyzing the AUC of the other three atherogenic risk indices evaluated at their high values (high total cholesterol/HDL-c, high triglycerides/HDL-c, High LDL-c/HDL-c), the results are close to 0.75 for the three formulas in both men and women, although they do not actually reach this figure, with the exception of the high triglycerides/HDL-c index in women, which reaches 0.76. These figures lead the test results to be considered average.

The ECORE-BF value to determine the presence of high values of the different atherogenic risk scales show an AUC around 0.8 for AD and LT and is slightly higher than 0.7 for the other three atherogenic indices, with somewhat higher values in men except for the triglyceride/HDL-C index.

As our results are divided into two groups of atherogenic risk index formulas—on the one hand, atherogenic dyslipidemia and lipid triad, and on the other, the high total cholesterol/HDL-c, high triglycerides/HDL-c, and high LDL-c indices/HDL-c—we wanted to check the degree of agreement between the measurements of the different formulas used. For this purpose, we carried out a Cohen’s kappa index, finding that between atherogenic dyslipidemia and lipid triad the result is 0.612, which is interpreted as a good agreement. Among the rest of the indices used, however, the concordance index is insignificant. Thus, the different formulas for calculating atherogenic risk are not interchangeable or substitutable with each other, although all of them have been validated.

The ECORE-BF index can be useful to alert us about a person’s atherogenic risk, since it presents an AUC greater than 0.7 with all the atherogenic risk indices used, and these results are valid in both men and women.

## 5. Strengths and Limitations

Among the study’s strengths, the substantial sample size, exceeding 386,000 individuals, provides a robust foundation for extrapolating findings to a broad population. This large sample size is essential for enhancing statistical precision and result reliability. The extensive analysis of five atherogenic risk scales further enhances understanding of the evaluated cardiovascular risk factors in the study. These strengths underscore the study’s robust design and the thoroughness of its analysis.

The origin of individuals from nine autonomous communities in Spain with a large population can be representative of the Spanish population. This suggests that data obtained from these communities may accurately reflect the demographic, social, or health characteristics of the entire Spanish population, thereby enhancing the generalizability and applicability of the results to a national context.

The agreement among the results of studies conducted in other countries and ethnicities regarding the association of obesity with atherogenic risk may enable us to extrapolate our findings to these populations. This highlights the importance and potential validity of the findings beyond the specific context of the original study.

The main limitation is that we do not have actual values of body fat, only an estimation based on a validated scale. The lack of direct measurement of body fat may introduce bias, as the scale used, although reliable, does not provide an exact measurement of the subjects’ body fat.

Another limitation is that this study excludes unemployed individuals, retirees, and individuals under 18 or over 69 years old. By excluding these groups, the results may not be generalizable to the entire population, as these groups could have characteristics or conditions that differ significantly from the subjects included in the study. However, given the sample size, we believe that excluding these groups does not significantly affect the results.

## 6. Conclusions

There is a good association between the risk scales: atherogenic dyslipidemia and lipid triad, and the ECORE-BF scale.

The ECORE-BF scale could be a useful and quick tool to screen atherogenic risk in primary care and occupational medicine consultations without the need for blood tests.

Due to its simplicity of application, the ECORE-BF scale is easy to implement in primary care and occupational medicine consultations, in order to establish preventive actions.

## Figures and Tables

**Figure 1 nutrients-16-02434-f001:**
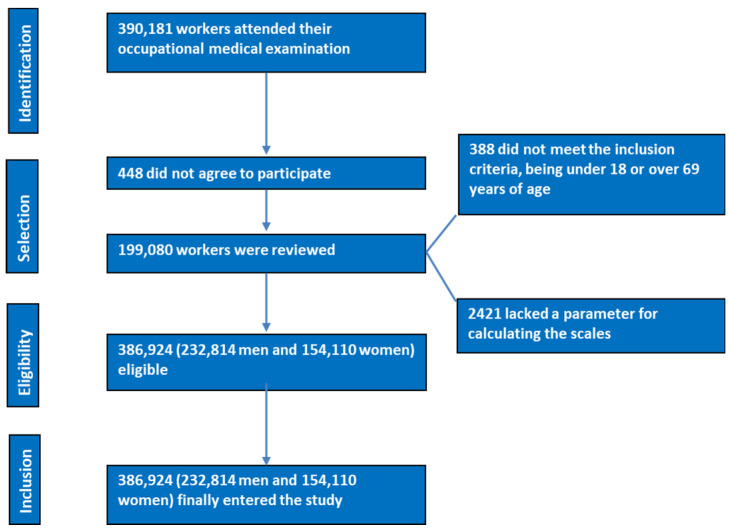
PRISMA flowchart of participants in the study.

**Figure 2 nutrients-16-02434-f002:**
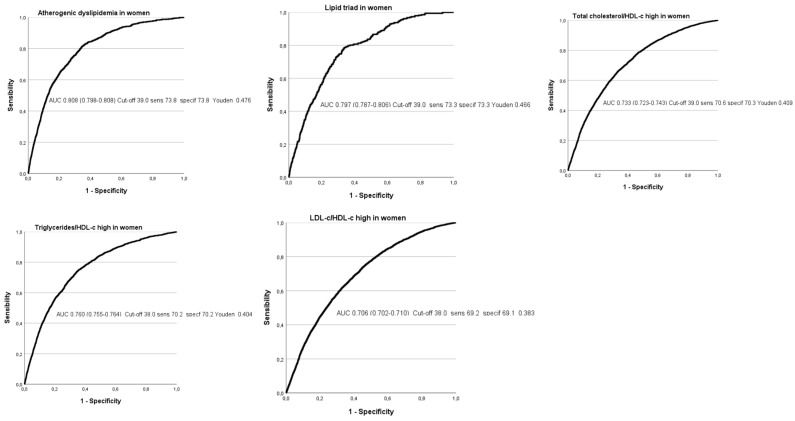
ROC curves of the atherogenic risk scales in women.

**Figure 3 nutrients-16-02434-f003:**
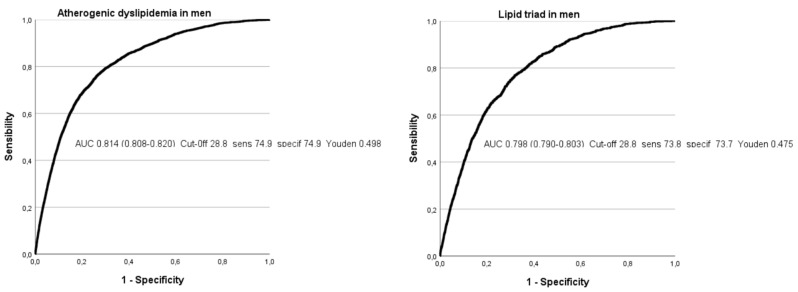
ROC curves of the atherogenic risk scales in men.

**Table 1 nutrients-16-02434-t001:** Characteristics of the population.

	Men *n* = 232,814	Women *n* = 154,110	
	Mean (SD)	Mean (SD)	*p*-Value (*t*-Student)
Age (years)	39.8 (10.3)	39.2 (10.2)	<0.001
Height (cm)	173.9 (7.0)	161.2 (6.6)	<0.001
Weight (kg)	81.1 (13.9)	65.3 (13.2)	<0.001
Waist circumference (cm)	87.7 (9.1)	73.9 (7.9)	<0.001
Hip circumference (cm)	100.0 (8.4)	97.2 (8.9)	<0.001
Systolic blood pressure (mmHg)	124.4 (15.1)	114.4 (14.8)	<0.001
Diastolic blood pressure (mmHg)	75.4 (10.6)	69.7 (10.3)	<0.001
Total cholesterol (mg/dL)	195.9 (38.9)	193.6 (36.4)	<0.001
HDL-c (mg/dL)	51.0 (7.0)	53.7 (7.6)	<0.001
LDL-c (mg/dL)	120.5 (37.6)	122.3 (37.0)	<0.001
Triglycerides (mg/dL)	123.8 (88.0)	88.1 (46.2)	<0.001
Glycaemia (mg/dL)	88.1 (12.9)	84.1 (11.5)	<0.001
	**%**	**%**	***p*-Value (Chi-Square)**
20–29 years	17.9	19.5	<0.001
30–39 years	33.1	33.3	
40–49 years	29.7	29.4	
50–59 years	16.3	15.3	
60–69 years	3.0	2.5	
Primary school	61.2	51.8	<0.001
Secondary school	34.0	40.7	
University	4.8	7.5	
Social class I	5.3	7.2	<0.001
Social class II	17.4	33.2	
Social class III	77.3	59.8	
No physical activity	54.5	47.8	<0.001
Yes physical activity	45.5	52.2	
Non Mediterranean diet	59.0	48.6	<0.001
Yes Mediterranean diet	41.0	51.4	
Non-smokers	62.9	67.0	<0.001
Smokers	37.1	33.0	

HDL-c: High density lipoprotein cholesterol. LDL-c: Low density lipoprotein cholesterol.

**Table 2 nutrients-16-02434-t002:** Mean values of ECORE-BF according to values of the atherogenic risk scales by gender.

		Men			Women	
	*n*	Mean (SD)	*p*-Value (*t*-Student)	*n*	Mean (SD)	*p*-Value (*t*-Student)
Non AD	221,582	25.4 (5.9)	<0.001	150,406	34.8 (6.9)	<0.001
AD	11,232	29.5 (5.4)		3704	41.1 (6.6)	
Non LT	228,166	25.5 (5.9)	<0.001	152,030	34.9 (7.0)	<0.001
LT	4648	28.7 (5.2)		2080	40.7 (6.6)	
Low TC/HDL-c	199,038	25.1 (5.9)	<0.001	122,908	34.3 (6.9)	<0.001
Moderate TC/HDL-c	33,182	28.5 (5.3)		301,54	37.7 (6.7)	
High TC/HDL-c	594	29.4 (4.5)		1048	38.8 (6.2)	
Low LDL-c/HL-c	176,954	24.9 (5.9)	<0.001	119,746	34.3 (7.0)	<0.001
High LDL-c/HL-c	55,860	27.7 (5.3)		34,364	37.3 (6.6)	
Low triglycerides/HDL-c	175,926	24.6 (5.7)	<0.001	141,372	34.5 (6.8)	<0.001
High triglycerides/HDL-c	56,888	28.6 (5.4)		12,738	40.2 (6.9)	

AD: Atherogenic dyslipidemia. LT: Lipid triad. TC: Total cholesterol. HDL-c: High density lipoprotein cholesterol. LDL-c: Low density lipoprotein cholesterol.

**Table 3 nutrients-16-02434-t003:** Prevalence of obesity with ECORE-BF criteria according to values of atherogenic risk scales by gender.

		Men			Women	
	*n*	%	*p*-Value (Chi-Square)	*n*	%	*p*-Value (Chi-Square)
Non AD	221,582	51.4	<0.001	150,406	44.2	<0.001
AD	11,232	79.6		3704	81.4	
Non LT	228,166	52.3	<0.001	152,030	44.6	<0.001
LT	4648	75.0		2080	79.6	
Low TC/HDL-c	199,038	48.9	<0.001	122,908	40.6	<0.001
Moderate TC/HDL-c	33,182	75.3		30,154	62.4	
High TC/HDL-c	594	83.8		1048	71.8	
Low LDL-c/HL-c	176,954	47.6	<0.001	119,746	40.8	<0.001
High LDL-c/HL-c	55,860	69.1		34,364	60.1	
Low triglycerides/HDL-c	175,926	45.6	<0.001	141,372	42.3	<0.001
High triglycerides/HDL-c	56,888	74.9		12,738	76.8	

AD: Atherogenic dyslipidemia. LT: Lipid triad. TC: Total cholesterol. HDL-c: High density lipoprotein cholesterol. LDL-c: Low density lipoprotein cholesterol.

**Table 4 nutrients-16-02434-t004:** Areas under the curve and cut-off points of ECORE-BF with its sensitivity, specificity, and Youden index in both genders.

Women	AUC (95% CI)	Cut-off-Sensitivity-Specificity-Youden
Atherogenic dyslipidemia	0.803 (0.798–0.808)	39.0-73.8-73.8-0.476
Lipid triad	0.797 (0.787–0.806)	39.1-73.3-73.3-0.466
High total cholesterol/HDL-c	0.733 (0.723–0.743)	39.0-70.3-70.6-0.414
High triglycerides/HDL-c	0.760 (0.755–0.764)	38.0-70.2-70.2-0.404
High LDL-c/HDL-c	0.706 (0.702–0.710)	38.0-69.2-69.1-0.383
**Men**	**AUC (95% CI)**	**Cut-off-Sensitivity-Specificity-Youden**
Atherogenic dyslipidemia	0.812 (0.809–0.815)	28.7-74.8-74.8-0.496
Lipid triad	0.798 (0.792–0.803)	28.8-73.8-73.7-0.475
High total cholesterol/HDL-c	0.744 (0.720–0.769)	28.1-70.9-70.9-0.418
High triglycerides/HDL-c	0.743 (0.741–0.745)	28.1-70.8-70.8-0.416
High LDL-c/HDL-c	0.716 (0.714–0.719)	28.0-69.9-69.8-0.397

HDL-c: High density lipoprotein cholesterol. LDL-c: Low density lipoprotein cholesterol.

**Table 5 nutrients-16-02434-t005:** Kappa Cohen’s indices.

	Atherogenic Dyslipidemia	Lipid Triad	High Total Cholesterol/HDL-c	High LDL-c/HDL-c	High Triglycerides/HDL-c
Atherogenic dyslipidemia	1	0.612	0.102	0.183	0.309
Lipid triad		1	0.182	0.110	0.149
High total cholesterol/HDL-c			1	0.026	0.035
High LDL-c/HDL-c				1	0.204
High triglycerides/HDL-c					1

HDL-c: High density lipoprotein cholesterol. LDL-c: Low density lipoprotein cholesterol.

## Data Availability

This study’s data are stored in a database that complies with all security measures at ADEMA-Escuela Universitaria. The Data Protection Delegate is Ángel Arturo López González.

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
