# Peer review of "Usefulness of the ECORE-BF Scale to Determine Atherogenic Risk in 386,924 Spanish Workers"

_nutrients, 2024, doi:10.3390/nu16152434_

Round 1

Reviewer 1 Report

Comments and Suggestions for Authors

Thank you for possibility to revire this interesting paper concerning ECORE BF score in determing atheromatic risk. 

my comments

1. The introduction is interesting but too long. I suggest moving some parts of this section to the discussion. 

2. What about patients with arterial hypertension treated medically and well-controlled? Those would present with normal blood pressure measurements. The same concerns lipid disorders and their treatment. What were the exclusion criteria? 

3. PLease present exact p values 

4. please correct misspellings (eg . fig 3)

5. In the discussion the authors mention ROC analysis and cut-off values as close to 0.8 while it seems that the authors mean AUC - please clarify 

6. Please add explanations for ROC analyses - what the authors meant to reveal? prediction of ECORE BF by other indices? 

Author Response

Dear reviewer,

First of all, thank you for your work and all your recommendations.

To facilitate your review, we have written the modifications in red in the article.

Comments and Suggestions for Authors

Thank you for possibility to revire this interesting paper concerning ECORE BF score in determing atheromatic risk. 

my comments

  1. The introduction is interesting but too long. I suggest moving some parts of this section to the discussion. 

Following your recommendation, we have shortened the introduction and moved part of this section to the discussion. Additionally, we have expanded the discussion by commenting on five articles, as advised by reviewer 2. We appreciate your suggestion.

  1. What about patients with arterial hypertension treated medically and well-controlled? Those would present with normal blood pressure measurements. The same concerns lipid disorders and their treatment. What were the exclusion criteria? 

Individuals were considered hypertensive or dyslipidemic if they were undergoing treatment, even if their values were normal.

The exclusion criteria were:

  • Age under 18 years or over 69 years.
  • Not being an employee of one of the participating companies.
  • Refusal to participate in the research study.
  • Refusal to consent to the use of data for epidemiological research.

For the calculation of atherogenic risk, the lipid levels of the patients were used regardless of whether they were undergoing treatment or not.

  1. PLease present exact p values 

The exact p-values are provided in each of the tables.

  1. please correct misspellings (eg . fig 3)

The spelling errors have been corrected. Thank you very much for your observation.

  1. In the discussion the authors mention ROC analysis and cut-off values as close to 0.8 while it seems that the authors mean AUC - please clarify 

The values close to 0.8, as you correctly noted, refer to the AUC and have been updated accordingly.

  1. Please add explanations for ROC analyses - what the authors meant to reveal? prediction of ECORE BF by other indices? 

What we intended to highlight is the value of ECORE-BF in predicting elevated values across all atherogenic risk scales (atherogenic indices, lipid triad, and atherogenic indices). We have added explanations of these data in the discussion section of the paper.

Thank you very much for your suggestions. We have addressed all of them and trust that our responses will adequately resolve your concerns.

Reviewer 2 Report

Comments and Suggestions for Authors

The quality of the figures and tables should be improved, including informative aspects for the Readers; professional assistance should be sought.

The authors conducted an observational, descriptive, and cross-sectional study involving 386,924 workers from various autonomous communities in Spain. They evaluated the association between the ECORE-BF scale and five atherogenic risk indices. The Authors observed that as atherogenic risk increased, the prevalence of obesity also rose, concluding that there is a significant association between atherogenic risk scales, including atherogenic dyslipidemia and the lipid triad, and the ECORE-BF scale. They suggested that the ECORE-BF scale could serve as a useful and quick tool for evaluating atherogenic risk in primary care and occupational medicine consultations without requiring blood tests.

The smoking status and drinking status must be included in the analysis distinguishing current, former, never (alongside medical treatments).

ROC Curves: please calculate the Youden’s index (quote PMID: 32964011, PMID:31771147) and report the highest J value on the Figures.

I am left underwhelmed by the discussion in its present form, which fails to interpret the data in the context of what is known in the field: it sounds somehow redundant, as it largely summarizes again data already presented in the Results without placing them in the proper scientific context. The following reports should be mentioned/discussed:

doi: 10.3390/nu11112833.

doi: 10.3390/nu10121912.

doi: 10.3390/jcm8122061.

doi: 10.1016/j.atherosclerosis.2023.117191.

doi: 10.1186/s12933-024-02175-5.

The strengths and limitations of the study should be deeply addressed, taking into account sources of potential bias or imprecision: Discuss both direction and magnitude of any potential bias.

Some of the conclusions drawn from the data seem to be overstated and should be toned down.

"Triglyverides"?

Comments on the Quality of English Language

-

Author Response

Dear reviewer,

First of all, thank you for your work and all your recommendations.

To facilitate your review, we have written the modifications in red in the article.

Comments and Suggestions for Authors

The quality of the figures and tables should be improved, including informative aspects for the Readers; professional assistance should be sought.

The authors conducted an observational, descriptive, and cross-sectional study involving 386,924 workers from various autonomous communities in Spain. They evaluated the association between the ECORE-BF scale and five atherogenic risk indices. The Authors observed that as atherogenic risk increased, the prevalence of obesity also rose, concluding that there is a significant association between atherogenic risk scales, including atherogenic dyslipidemia and the lipid triad, and the ECORE-BF scale. They suggested that the ECORE-BF scale could serve as a useful and quick tool for evaluating atherogenic risk in primary care and occupational medicine consultations without requiring blood tests.

The smoking status and drinking status must be included in the analysis distinguishing current, former, never (alongside medical treatments).

Unfortunately, we do not have the data you requested. Our database does not contain information on the treatments the patients are undergoing or their alcohol consumption history (current, former, or never). Additionally, the data are dissociated, meaning we would need to re-encode the patient key and manually access each patient's medical records to gather the requested information, which is practically impossible given our sample size of 386,924 patients. Furthermore, these factors are not part of the formula for the ECORE-BF scale or the atherogenic risk scales, so we believe they would not alter the results. We apologize for not being able to provide this data. Thank you very much.

ROC Curves: please calculate the Youden’s index (quote PMID: 32964011, PMID:31771147) and report the highest J value on the Figures.

The Youden index is shown in Table 4. Following your recommendations, we have included it alongside the highest J-value in the figures. Thank you very much.

I am left underwhelmed by the discussion in its present form, which fails to interpret the data in the context of what is known in the field: it sounds somehow redundant, as it largely summarizes again data already presented in the Results without placing them in the proper scientific context. The following reports should be mentioned/discussed:

doi: 10.3390/nu11112833.

doi: 10.3390/nu10121912.

doi: 10.3390/jcm8122061.

doi: 10.1016/j.atherosclerosis.2023.117191.

doi: 10.1186/s12933-024-02175-5.

Following your recommendations, we have expanded the discussion by commenting on the five articles you suggested. Additionally, in accordance with reviewer 1's recommendations, we have shortened the introduction and moved part of this section to the discussion. We appreciate your suggestion.

The strengths and limitations of the study should be deeply addressed, taking into account sources of potential bias or imprecision: Discuss both direction and magnitude of any potential bias.

We have thoroughly discussed the strengths and limitations of the study, detailing the potential biases. Thank you very much.

Some of the conclusions drawn from the data seem to be overstated and should be toned down.

We have revised the study conclusions to be more measured, following your recommendation. Thank you very much.

"Triglyverides"?

We have made the changes in the ROC curve for women. Thank you for your observation.

Thank you very much for your suggestions. We have addressed all of them and trust that our responses will adequately resolve your concerns.

Reviewer 3 Report

Comments and Suggestions for Authors

The study is based on a significant sample size. I recommend making a few changes.

The results in the submitted abstract are described superficially - too much methodology and too few specific data. It is unclear how significant the obtained results are.

„The chi-square test and Student's t test, provided the samples were independent, were employed to assess the relationship between the various variables” - Please specify the purpose of the applied tests. The Student's t-test is not used to examine relationships. The fulfillment of the assumptions necessary to apply the indicated tests should be described

In the results, it is not clear where and which statistical test was applied.

For the Student's t-test and chi-square test, it is recommended to calculate the appropriate measure of effect size to show the practical significance of the obtained results (Cohen’s D and phi/Cramer’s V).

The fulfillment of the main assumptions of the applied regression analysis should be described.

Comments on the Quality of English Language

Minor editing of English language required.

Author Response

Dear reviewer,

First of all, thank you for your work and all your recommendations.

To facilitate your review, we have written the modifications in red in the article.

Comments and Suggestions for Authors

The study is based on a significant sample size. I recommend making a few changes.The results in the submitted abstract are described superficially - too much methodology and too few specific data. It is unclear how significant the obtained results are.

Thank you for your observation. We have provided a more detailed description of the results in the abstract.

„The chi-square test and Student's t test, provided the samples were independent, were employed to assess the relationship between the various variables” - Please specify the purpose of the applied tests.

The Chi-square test was applied to assess the differences between frequencies, while the t-test was used to evaluate the differences between means.

The Student's t-test is not used to examine relationships. The fulfillment of the assumptions necessary to apply the indicated tests should be described

You are absolutely correct; it was a terminological error, as the t-test does not measure relationships but only determines if there are statistically significant differences between two variables. To apply the statistical tests, validation tests were conducted beforehand to confirm that the sample follows a normal distribution, which allows for the subsequent use of parametric statistical tests. We have proceeded to specify this.Thank you very much.

In the results, it is not clear where and which statistical test was applied.

Thank you for your observation. We have detailed the statistical test applied in each table.

For the Student's t-test and chi-square test, it is recommended to calculate the appropriate measure of effect size to show the practical significance of the obtained results (Cohen’s D and phi/Cramer’s V).

As previously mentioned, validation tests were conducted prior to the use of the Chi-square test and the t-test. We have proceeded to specify this.

The fulfillment of the main assumptions of the applied regression analysis should be described.

Thank you for your observation. A multinomial logistic regression analysis was not conducted, as incorrectly indicated in the statistical analysis section. We have corrected this.

Comments on the Quality of English Language

Minor editing of English language required.

The text has been reviewed and corrected by Meryl Jons, professional translator of medical manuscripts, Academia WYN in Mallorca. We trust that all defects have been corrected. Thank you so much.

Thank you very much for your suggestions. We have addressed all of them and trust that our responses will adequately resolve your concerns.

Round 2

Reviewer 2 Report

Comments and Suggestions for Authors

-

Comments on the Quality of English Language

-

Reviewer 3 Report

Comments and Suggestions for Authors

The recommended changes have been incorporated.

Comments on the Quality of English Language

Minor editing of English language required.